JCB | Journal of Cell Biology

# Structural basis for the nuclear import and export functions of the biportin Pdr6/Kap122

Metin Aksu, Sergei Trakhanov, Arturo Vera Rodriguez, and Dirk Görlich

**Importins ferry proteins into nuclei while exportins carry cargoes to the cytoplasm. In the accompanying paper in this issue (Vera Rodriguez et al. 2019. *J. Cell Biol.* https://doi.org/10.1083/jcb.201812091), we discovered that Pdr6 is a biportin that imports, e.g., the SUMO E2 ligase Ubc9 while depleting the translation factor eIF5A from the nuclear compartment. In this paper, we report the structures of key transport intermediates, namely, of the Ubc9•Pdr6 import complex, of the RanGTP•Pdr6 heterodimer, and of the trimeric RanGTP•Pdr6•eIF5A export complex. These revealed nonlinear transport signals, chaperone-like interactions, and how the RanGTPase system drives Pdr6 to transport Ubc9 and eIF5A in opposite directions. The structures also provide unexpected insights into the evolution of transport selectivity. Specifically, they show that recognition of Ubc9 by Pdr6 differs fundamentally from that of the human Ubc9-importer Importin 13. Likewise, Pdr6 recognizes eIF5A in a nonhomologous manner compared with the mammalian eIF5A-exporter Exportin 4. This suggests that the import of Ubc9 and active nuclear exclusion of eIF5A evolved in different eukaryotic lineages more than once and independently from each other.**

## Introduction

Members of the importin β superfamily function as nuclear transport receptors (NTRs) to carry cargoes through the permeability barrier of nuclear pore complexes (NPCs). The superfamily includes mediators of import into the nucleus (called importins) as well as exportins. The RanGTPase system supplies energy for and governs directionality of active transport (Görlich and Kutay, 1999; Christie et al., 2016; Matsuura, 2016).

Importins capture cargo at low RanGTP levels in the cytoplasm and release cargo upon RanGTP-binding into the nucleus before returning to the cytoplasm for Ran release/GTP hydrolysis and loading another cargo. Exportins operate the other way around, carrying cargo in their RanGTP-bound state out of nuclei. NTR interactions with RanGTP and cargo are, therefore, antagonistic in the case of importins but synergistic in the case of exportins.

The yeast *Saccharomyces cerevisiae* possesses 14 importin β-type NTRs and humans possess an even 20 (Fornerod et al., 1997; Görlich et al., 1997). Most NTRs function either as an importin or an exportin, for example Importin β itself or CRM1/Exportin 1/Xpo1 (Christie et al., 2016; Matsuura, 2016). A few, however, act as "biportins" and carry one set of cargoes into and another set out of nuclei (Kaffman et al., 1998; Lipowsky et al., 2000; Mingot et al., 2001, 2004; Yoshida and Blobel, 2001; Gontan et al., 2009; Aksu et al., 2018).

NTRs share only little sequence identity with each other (typically <15% between paralogues); however, they are all structurally related and made up of HEAT repeats (Görlich et al., 1997; Chook and Blobel, 1999; Cingolani et al., 1999; Vetter et al., 1999). HEAT repeats are ∼40–amino acid motifs, which consist of two consecutive α-helices (A and B) that pack in an antiparallel orientation against each other. Individual repeats, in turn, pack side by side to form a right-handed superhelical structure; the A helices form the outer surface, and the B helices form the inner surface. Such an arrangement of repeats gives plasticity to the receptors, which indeed adopt a variety of conformations, e.g., from a closed toroid to an open solenoid to bind their cargoes and/or respond to RanGTP (Chook and Blobel, 1999; Cingolani et al., 1999; Vetter et al., 1999; Matsuura and Stewart, 2004; Cook et al., 2005, 2009; Lee et al., 2005; Dong et al., 2009; Monecke et al., 2009, 2013; Bono et al., 2010; Grünwald and Bono, 2011; Aksu et al., 2016).

NTRs recognize their cargoes through transport signals, which in the simplest case represent linear peptide motifs. Xpo1, for example, is recruited by leucine-rich nuclear export signals (NESs), which have a length of just 9 to ∼15 amino acids and comprise four to five characteristically spaced hydrophobic residues that dock into dedicated binding pockets of the exportin (Dong et al., 2009; Monecke et al., 2009; Güttler et al., 2010; Fung et al., 2015). NESs typically reside in disordered C- or

Department of Cellular Logistics, Max Planck Institute for Biophysical Chemistry, Göttingen, Germany.

Correspondence to Dirk Görlich: goerlich@mpibpc.mpg.de; M. Aksu's present address is Department of Biochemistry, Oxford University, Oxford, UK.

Rockefeller University Press
J. Cell Biol. 2019 Vol. 218 No. 6 1839–1852

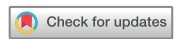

https://doi.org/10.1083/jcb.201812093 1839

N-terminal extensions of a protein and can easily be transplanted from one protein to another. This may explain why so many proteins are exported by Xpo1.

Classic or canonical nuclear localization signals (NLSs) also function as linear motifs. They comprise either one or two short clusters of basic residues (Kalderon et al., 1984; Robbins et al., 1991) that dock into cognate binding pockets of the nuclear import adapter Importin α, which in turn uses Importin β as the actual transport receptor (Görlich et al., 1994, 1995; Imamoto et al., 1995; Conti et al., 1998; Cingolani et al., 1999).

However, we also know of complex and three-dimensional nuclear transport signals. These are typically associated with a chaperone function of the NTRs. Xpo2/CAS/Cse1, for example, exports Importin α in an autoinhibited state where the NLS-binding site is occluded, thereby preventing an NLS-dependent reexport of previously imported nuclear proteins (Kutay et al., 1997; Matsuura and Stewart, 2004). Xpo2, therefore, recognizes the fold and even a specific conformation of Importin α.

A similar principle applies to the nuclear export of the translation elongation factor eIF5A (Lipowsky et al., 2000; Aksu et al., 2016), which is required for the efficient synthesis of polyproline stretches (Doerfel et al., 2013; Gutierrez et al., 2013; Ude et al., 2013). eIF5A comprises two globular domains (Tong et al., 2009) and contains a unique, twofold positively charged amino acid, called hypusine, that is essential for eIF5A function, as well as for cell viability (Shiba et al., 1971; Schnier et al., 1991). Due to its small size (17 kD), eIF5A readily leaks through the sieve-like barrier of NPCs into nuclei (Lipowsky et al., 2000) where it is not only lost for its cytoplasmic function but also might even engage in deleterious off-target interactions, such as nonspecific RNA binding or competition with the ribosome export-adapter Nmd3 (Malyutin et al., 2017).

The mammalian Exportin Xpo4 captures such mislocalized nuclear eIF5A and retrieves it back to the cytoplasm (Lipowsky et al., 2000). The structure of the corresponding export complex revealed that Xpo4 does not recognize a linear sequence (Aksu et al., 2016). Instead, intra-repeat loops of Xpo4's HEAT repeats contact both eIF5A domains, including the essential hypusine, and shield the 25S RNA- and tRNA-binding interface (Aksu et al., 2016; Melnikov et al., 2016; Schmidt et al., 2016). Therefore, Xpo4 acts like a compartment-specific antagonist of eIF5A function and as a suppressor of off-target interactions. Xpo4 was also shown to import Sox-type transcription factors into the nucleus and, thus, to function as a biportin (Gontan et al., 2009). Meanwhile, numerous additional Xpo4 cargoes have been identified (Kimura et al., 2017; Mackmull et al., 2017).

A recognition of a globular domain as a nuclear transport signal was also observed for certain import cargoes, Ubc9 being a prominent example. Ubc9 is an essential, predominantly nuclear SUMO-conjugating E2 enzyme (Seufert et al., 1995; Gong et al., 1997; Lee et al., 1998; Flotho and Melchior, 2013). Mammalian Ubc9 is imported by importin 13/Imp13, which is encoded by the Ipo13 gene (Mingot et al., 2001). The structure of Ubc9 shows a canonical E2 catalytic fold formed from a single α + β structure domain (Tong et al., 1997). In the Imp13•Ubc9 import complex, Imp13 encloses Ubc9, in particular via HEAT repeats 1–9

(Grünwald and Bono, 2011). This masks most of the Ubc9's interacting residues with other proteins, such as with E1 and E3 SUMO-conjugating enzymes, as well as with potential conjugation substrates (Wang et al., 2010; Grünwald and Bono, 2011; Gareau et al., 2012; Streich and Lima, 2016; see Protein Data Bank [PDB] 2XWU, 3ONG, 3UIN, and 5JNE). Imp13 is, therefore, expected to block Ubc9 activity during transport. In addition to import, Imp13 also functions in export, depleting eIF1A and other proteins from the cell nucleus (Mingot et al., 2001; Grünwald et al., 2013; Baade et al., 2018). It is, thus, a biportin as well.

*S. cerevisiae* Pdr6/Kap122 was originally identified as a pleiotropic drug–resistance protein (Chen et al., 1991) and subsequently as an importin β superfamily member (Görlich et al., 1997). It has been shown to account for nuclear import of TFIIA (Titov and Blobel, 1999) and a ribonucleotide reductase complex with Wtm1 (Zhang et al., 2006). In the accompanying paper in this issue (Vera Rodriguez et al.), we discovered that Pdr6 is actually a biportin (Fig. 1), which brings it into the same category as Imp13 and Xpo4. Furthermore, we found that Pdr6 imports Ubc9, exports eIF5A, and thus, combines functions of Imp13 and Xpo4.

In this study, we elucidated the structures of Pdr6 in complex with Ubc9, with RanGTP, and with RanGTP and eIF5A. Pdr6 binds Ubc9 in an open superhelix conformation. This is similar to Imp13. Pdr6, however, captures Ubc9 in a different orientation, through different HEAT repeats (inner helices of HEATs 6–18), and through a different and only marginally overlapping set of Ubc9 residues. RanGTP releases Ubc9 by a partial blocking of Ubc9-interacting residues and enforcing ring closure of the Pdr6 superhelix. The structure of the RanGTP•Pdr6•eIF5A complex also revealed striking differences in cargo recognition compared with Xpo4. Unlike Xpo4, which attracts eIF5A through its toroid surface, Pdr6 clamps eIF5A between the N- and C-terminal halves and contacts this cargo mainly through the inter-repeat loops of HEATs 8–11. In summary, this suggests that the here-described cargo interactions of Pdr6 are not homologous to the mammalian Ubc9 and eIF5A transporters, that noncanonical import of Ubc9 and export of eIF5A appeared at least twice in evolution, that there is evolutionary pressure to transport these cargoes in a chaperoned manner, and that the equipment of eukaryotic species with NTRs is more plastic than previously thought.

## Results

To obtain insights into the molecular transport mechanisms mediated by Pdr6, we decided to crystallize key transport intermediates (see Fig. 1 for a scheme) and determine their structures. To this end, we recombinantly expressed and purified complex constituents, performed pretrials to assemble individual complexes, optimized complex constituents, and finally started robotic crystallization screens and manual refinements of crystallization conditions.

### Structure determination of the Pdr6 complexes

As a first hit, we obtained hexagonal crystals for a complex of full-length Pdr6 and the GTPase-deficient human RanQ69L

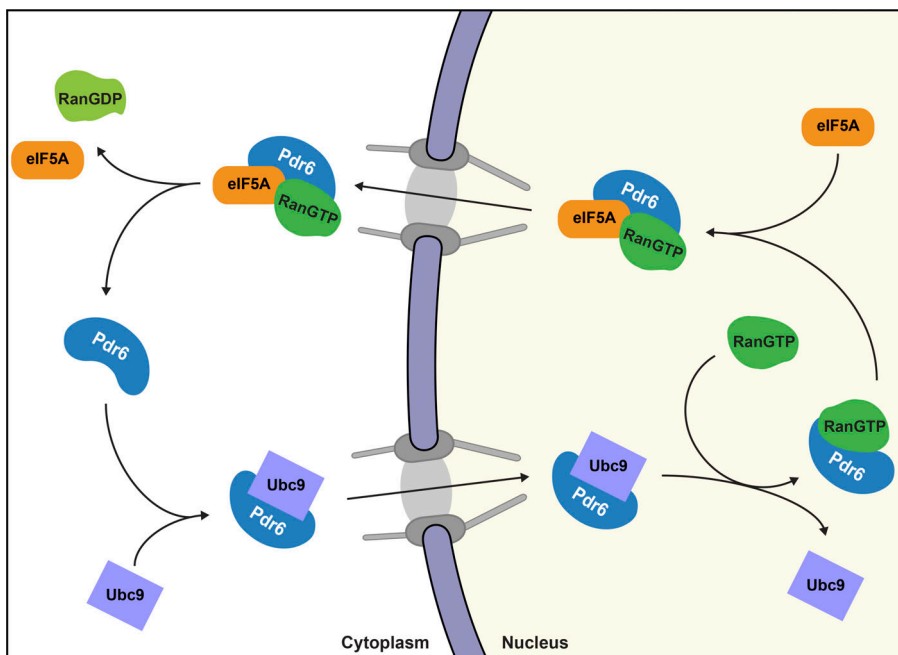

Figure 1. **Schematic illustration of Pdr6's nucleocytoplasmic transport cycle.** Pdr6 is a biportin. It has several alternative import and export cargoes; however, only Ubc9 and eIF5A are shown.

mutant (Bischoff et al., 1994; Klebe et al., 1995) comprising residues 5–180 and, thus, lacking its auto-inhibitory C-terminus (Richards et al., 1995). The best one diffracted to 3 Å. The structure was solved by a combination of molecular replacement (MR) using human Ran (PDB 3GJX; Monecke et al., 2009) as a search model and single-wavelength anomalous dispersion phasing (Hendrickson et al., 1990) on a dataset with selenomethionine-substituted Pdr6. The final model was refined to an $R_{work}$ of 22% and an $R_{free}$ of 25% (Table 1). It comprised Ran residues 8–178 and Pdr6 residues 3–1,076 with only a few residues missing from a number of loops that showed no electron density. The structure is described below.

The crystallized export complex included the Pdr6•RanGTP subcomplex and *S. cerevisiae* eIF5A lacking its 15 N-terminal disordered residues. Given that hypusine is critical for the eIF5A–Xpo4 interaction (Aksu et al., 2016), it was rather surprising that hypusination of eIF5A had no effect on complex formation. eIF5A was, therefore, used in the nonmodified form.

We eventually obtained hexagonal crystals of the ternary RanGTP•Pdr6•eIF5A complex that diffracted to 3.7 Å (Table 1). The structure was solved by MR by using as search models the binary Pdr6•RanGTP complex, as well as yeast eIF5A that had been homology-modeled with a human eIF5A template (PDB 5DLQ). The asymmetric unit contained two ternary complexes, whereby complex 1 (corresponding to chains A, B, and C) was slightly better defined. The final model was refined to an $R_{work}$ of 21% and an $R_{free}$ of 25% and comprised Ran residues 8–178, Pdr6 residues 3–1076, and eIF5A residues 17–153.

Diffracting Ubc9-bound Pdr6 crystals were obtained from a complex comprising full-length *S. cerevisiae* Ubc9 and Pdr6 proteins. The structure was solved by MR using Pdr6 (from the Pdr6•RanGTP complex) and Ubc9 (PDB 2GJD; van Waardenburg et al., 2006) as search models. A successful MR solution was obtained only after splitting Pdr6 into rigid bodies (see Materials

and methods). The final model was refined to 4.5-Å resolution with an $R_{work}$ of 26% and an $R_{free}$ of 27% (Table 1). Despite the low resolution, we obtained a reliable signal for almost the complete chains: the Ubc9 model included residues 2–157 while the Pdr6 model comprised residues 3–1,076 and lacked only an additional inter-repeat loop.

## Structure of the Pdr6•Ubc9 import complex

Pdr6 is an all α-helical protein built of 19 canonical HEAT repeats followed by three helices (termed HEAT 20) at its very C-terminal region (Fig. 2, A–D). The repeats generally pack side by side with a right-handed twist. HEAT 10 is kinked away from HEATs 9 and 11 and divides Pdr6 into N- and C-terminal arches. Helices within the HEAT repeats are mostly connected by short loops.

Most of the so-far analyzed importins and exportins (except for Imp13 and Xpo2) contain a large intra-repeat insertion or "acidic loop" at HEAT 8 or HEAT 9 that is important for cargo and/or Ran binding (Chook and Blobel, 1999; Cingolani et al., 1999; Vetter et al., 1999). Pdr6 lacks such a large insertion. It has, however, two small helices between the A and B helices of HEAT 4 that pack against HEATs 3–5. Moreover, Pdr6 contains several inter-repeat insertions, mostly in the form of small helices; two of them contact the export cargo and are discussed below.

In the import complex, Pdr6 adopts an open conformation (Fig. 2, C and D) where the distance between the tip of HEAT 20 and the loop between HEATs 4 and 5 is ~25 Å. Ubc9 is bound to the inner surface of the superhelix (Fig. 2, C and D) with HEATs 6–18 of Pdr6 engaging in extensive interactions with Ubc9 (Fig. 2 D and Fig. S1). The first interaction interface is formed by residues of HEATs 6–9 and involves mostly salt bridges. Several aspartic acid residues of Pdr6 at this region (D370, D429, D433, D436, D492, and D499) form a negatively charged patch,

Table 1. **Data collection and refinement statistics**

| | Complex | | |
| | Pdr6·RanGTP | Pdr6·Ubc9 | RanGTP·Pdr6·eIF5A |
|---|---|---|---|
| **PDB code** | 6Q82 | 6Q83 | 6Q84 |
| **Data collection** | | | |
| Space group | $P6_322$ | $R32$ | $P6_3$ |
| Cell dimensions | | | |
| $a, b, c$ (Å) | 193.0, 193.0, 229.2 | 198.3, 198.3, 289.6 | 139.5, 139.5, 346.6 |
| α, β, γ (°) | 90.0, 90.0, 120.0 | 90.0, 90.0, 120.0 | 90.0, 90.0, 120.0 |
| Resolution (Å) | 59.89–2.99 (3.10–2.99) | 49.56–4.53 (4.69–4.53) | 49.17–3.70 (3.83–3.70) |
| Unique reflections | 51,160 (4,876) | 12,933 (1,242) | 40,583 (4,014) |
| Completeness (%) | 99.7 (97.4) | 99.41 (96.5) | 99.83 (99.7) |
| $R_{merge}$ (%) | 27 (776) | 17 (242) | 9 (292) |
| $R_{pim}$ (%) | 1 (38) | 6 (83) | 3 (95) |
| $I/\sigma I$ | 44.70 (2.32) | 11.15 (0.95) | 17.74 (1.60) |
| CC1/2 | 100 (89.3) | 100 (58.5) | 100 (53.7) |
| Multiplicity | 399.8 (416.7) | 10.1 (9.4) | 10.8 (10.0) |
| **Refinement** | | | |
| Resolution (Å) | 59.89–2.99 | 49.56–4.53 | 49.17–3.70 |
| No. reflections | 51,056 (4,876) | 12,933 (1,238) | 40,554 (4,014) |
| $R_{work}$ (%) | 21.8 (31.4) | 25.9 (40.1) | 20.7 (32.1) |
| $R_{free}$ (%) | 25.3 (35.7) | 27.1 (36.6) | 24.7 (33.4) |
| No. atoms | | | |
| Protein | 9,649 | 9,380 | 21,141 |
| Ligand/ion | 33 | | 66 |
| $B$-factors | | | |
| Protein | 111 | 311 | 197 |
| Ligand/ion | 91 | | 179 |
| RMSDs | | | |
| Bond lengths (Å) | 0.003 | 0.005 | 0.002 |
| Bond angles (°) | 0.70 | 0.88 | 0.48 |
| MolProbity analysis | | | |
| Ramachandran favored (%) | 94.24 | 93.64 | 94.00 |
| Ramachandran outliers (%) | 0.59 | 0.52 | 0.54 |
| Clash score | 11.05 | 10.79 | 6.18 |
| MolProbity score | 1.95 | 2.29 | 1.74 |

Statistics for the highest-resolution shell are shown in parentheses.

shielding the conserved basic residues at the helix α1, as well as the following loop and two β-strands of Ubc9 (Fig. 2 E and Fig. S1).

The second set of interactions involves HEATs 11–18 of Pdr6. Helices α4 and α3 of Ubc9 pack against HEATs 12–15 and HEATs 16–17, respectively (Fig. S1). Hydrophobic and polar contacts govern the interactions at this interface. Among the interacting residues, W134[Ubc9], R135[Ubc9], L150[Ubc9], Q154[Ubc9], Y155[Ubc9], S156[Ubc9], and K157[Ubc9] of helix α3 constitute the nonconserved

residues and might be the site giving Pdr6 species specificity in terms of Ubc9 binding (see below).

Overall, Pdr6 buries 1,473 Å² of the Ubc9 surface area. This includes most of the Ubc9's E1 (PDB 5FQ2, 3ONG, and 4W5V), E3 (PDB 5JNE and 3UIN), and the substrate (PDBs 5JNE and 3UIN) interaction interfaces (Fig. 2 F; Wang et al., 2010; Gareau et al., 2012; Streich and Lima, 2016). Therefore, one would assume that Pdr6 acts as an inhibitor of Ubc9 and carries this SUMO E2 ligase in an inactive state. On the contrary, Pdr6 does not shield the

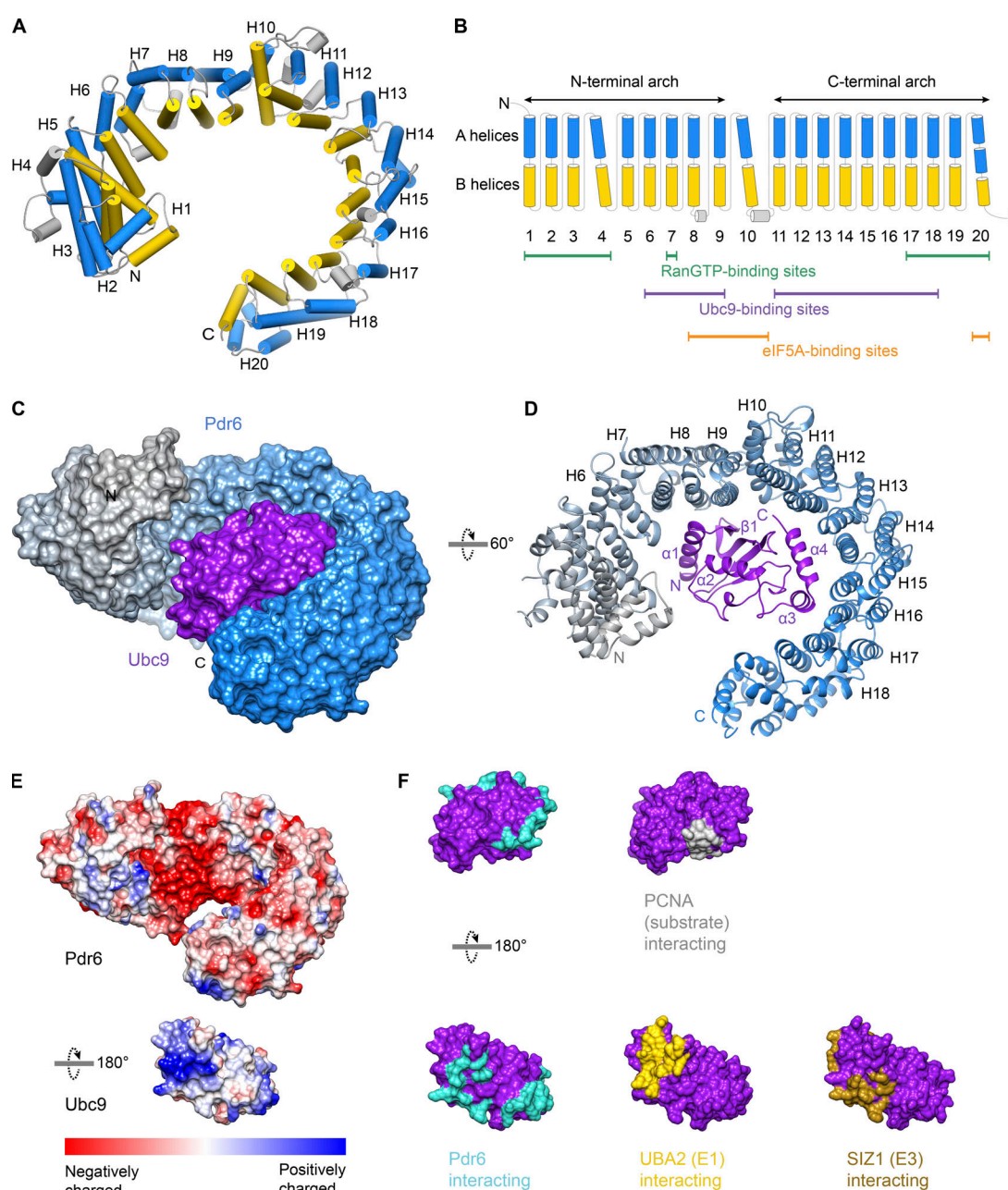

Figure 2. **Structure of the Pdr6·Ubc9 import complex and overview of Pdr6 secondary structure.** **(A)** Pdr6 in the import complex is shown. A and B helices of the HEAT repeats are represented as cylinders and shown in blue and yellow, respectively. **(B)** Schematic representation of the Pdr6 secondary structure. Coloring is the same as in A. Tilted cylinders represent interruptions in the right-handed arrangement of successive HEAT repeats. Green, purple, and orange lines below the cylinders indicate repeats that interact with RanGTP, Ubc9, and eIF5A, respectively. **(C and D)** View of the complex in two different orientations. **(C)** The import complex is shown as the surface representation. Pdr6 is depicted with a color gradient from gray (N-terminus) to blue (C-terminus) while Ubc9 is shown in purple. **(D)** The complex is rendered as a ribbon representation. The secondary structure of interacting Pdr6 and Ubc9 regions are labeled. **(E)** Pdr6 and Ubc9 surfaces are shaded according to electrostatic potential with a color gradient from red (negatively charged, −10 kcal/mol) to blue (positively charged, +10 kcal/mol). Pdr6 is shown in the same orientation as in C while Ubc9 is rotated 180°. **(F)** Ubc9 is rendered as a surface representation, shaded in purple and shown in two orientations. Interaction interfaces of Ubc9 are marked in cyan (Pdr6 contacts), yellow (UBA2 contacts, PDB 3ONG), brown (SIZ1 contacts, PDB 5JNW), or gray (PCNA contacts, PDB 5JNE).

SUMO-binding site of Ubc9. In fact, Ubc9 with a thioester-bonded SUMO would bind Pdr6 without a significant clash (PDB 5JNE and 3UIN), and a nearby lysine (K956) could even act as a SUMO-acceptor site (with an "inverted consensus motif"; Impens et al., 2014) and thus possibly make Pdr6 a SUMOylation substrate.

**The Pdr6·Ubc9 interaction is not homologous with its Imp13·Ubc9 counterpart**

The structure of Ubc9-bound Pdr6 resembles that of Imp13 (Fig. 3, A and B). Yet, the cargo-recognition mechanism differs substantially. First, while Ubc9 is recognized via the N-terminal arch of Imp13, the corresponding region of Pdr6 has no role in

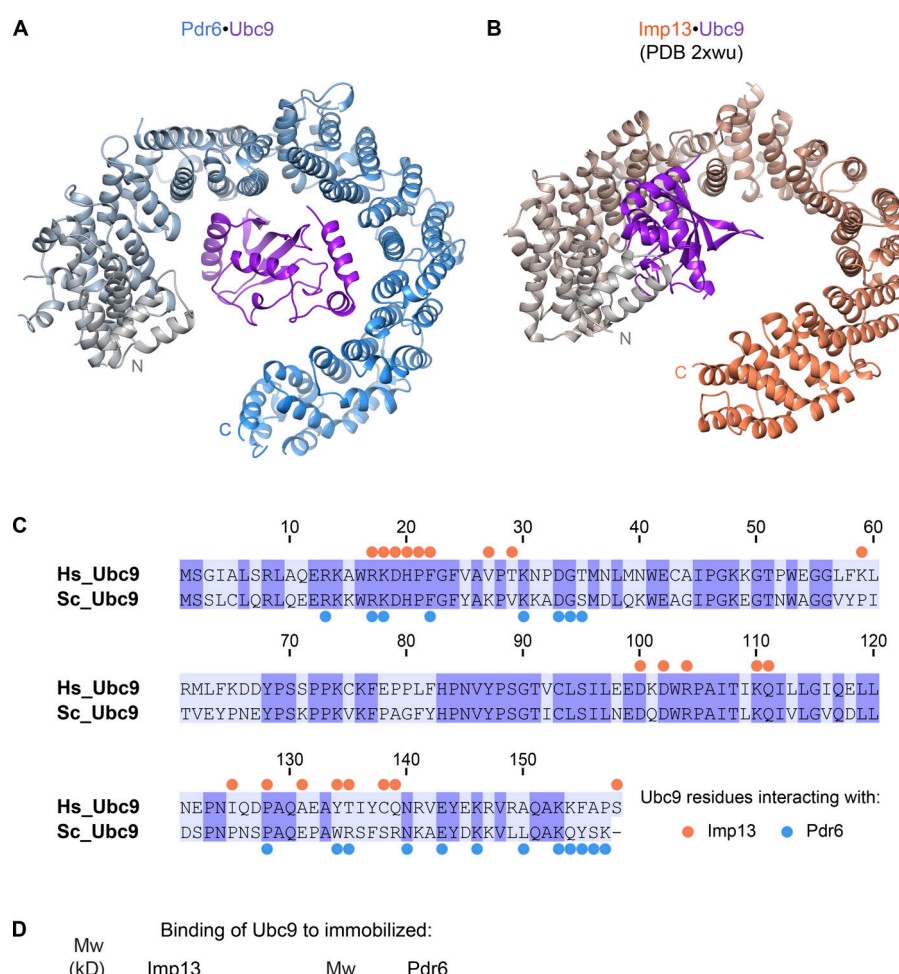

**A** Pdr6•Ubc9

**B** Imp13•Ubc9 (PDB 2xwu)

**C**

Hs_Ubc9 / Sc_Ubc9 sequence alignment (residues 1–160)

Ubc9 residues interacting with:
- Imp13 (orange)
- Pdr6 (blue)

**D** Binding of Ubc9 to immobilized:

Imp13 — 100, Imp13; 10, Ubc9 — Yeast, Human
Pdr6 — 100, Pdr6; 10, Ubc9 — Yeast, Human

Figure 3. **Comparison of import cargo recognition by Pdr6 and Imp13. (A and B)** Ribbon representations of Pdr6•Ubc9 (A) and Imp13•Ubc9 complexes (B) are shown. Corresponding Pdr6 and Imp13 structures are aligned with respect to the first three HEAT repeats of the NTRs and shown in the same orientation as in Fig. 2 D. Pdr6 is depicted with a color gradient from gray (N-terminus) to blue (C-terminus) while Imp13 is from gray to salmon. Ubc9 is shown in purple. **(C)** Sequence alignment of human and yeast Ubc9 is shown with identical residues shaded in purple. Blue and salmon dots represent Ubc9 residues that contact Pdr6 and Imp13, respectively. **(D)** H$_{14}$-ZZ-bdSUMO–tagged Pdr6 and Imp13 (1 µM) were incubated with human or yeast Ubc9 (2 µM). Formed import complexes were retrieved via tagged NTRs, eluted by (the tag-cleaving) bdSENP1 protease, and analyzed by SDS-PAGE and Coomassie blue staining.

Ubc9 binding (Fig. 3, A and B; Grünwald and Bono, 2011). Instead, C-terminal HEATs are involved in Ubc9 recognition. Second, Ubc9 residues interacting with either of the two NTRs only marginally overlap and include both conserved and non-conserved residues (Fig. 3 C). This also explains the species specificity of Pdr6 and Imp13 toward Ubc9 binding (Fig. 3 D). In summary, this makes it very unlikely that the Pdr6•Ubc9 and the Ubc9•Imp13 interactions are homologous to each other. Instead, it appears that nuclear import of Ubc9 in a chaperoned manner was "invented" at least twice during evolution.

**Structure of the Pdr6•RanGTP complex and mechanism of import cargo release into the nucleus**

RanGTP needs to replace the import cargo from NTR before the latter returns to the cytoplasm. An NTR•RanGTP complex can thus be considered an endpoint of such a displacement reaction. Our Pdr6•RanGTP structure is indeed incompatible with Ubc9 binding (Fig. 4). There is a partial overlap of the respective binding sites (Fig. 2 B), i.e., RanGTP would clash with a bound Ubc9 molecule. Furthermore, RanGTP changes the curvature of the Pdr6 superhelix from a rather open conformation to a closed

toroid. As a consequence, the binding surface loses its shape complementary to Ubc9 and then also clashes with the cargo.

The ring closure is not caused by a movement around a single hinge region but instead by subtle changes in the packing of adjacent helices all along the Pdr6 molecule until HEATS 19 and 20 contact the loop between HEATS 4 and 5. RanGTP binding clearly provides a driving force. It interacts with an N-terminal region of Pdr6 (HEATs 1–4 and 7), as well as with a C-terminal one (HEATs 17–20), and thereby pulls the NTR into a ring-like shape. As discussed below, we assume that Ran has to put the superhelical structure under tension, and this expenditure of energy could explain why the binary RanGTP•Pdr6 interaction is of only moderate affinity (∼230 nM; Hahn and Schlenstedt, 2011) even though a rather larger surface area of Ran (1,938 Å²) is buried. For comparison, importin β binds RanGTP with ∼1 nM affinity (Bischoff and Görlich, 1997).

The structure of Ran itself is essentially identical (root mean square deviation [RMSD], 1 Å) to that of other NTR•RanGTP complexes. Ran is positioned inside the Pdr6 toroid and, thus, enclosed by Pdr6's N- and C-terminal arches (Fig. 4, A and B). This topology is rather similar to the human Imp13•RanGTP

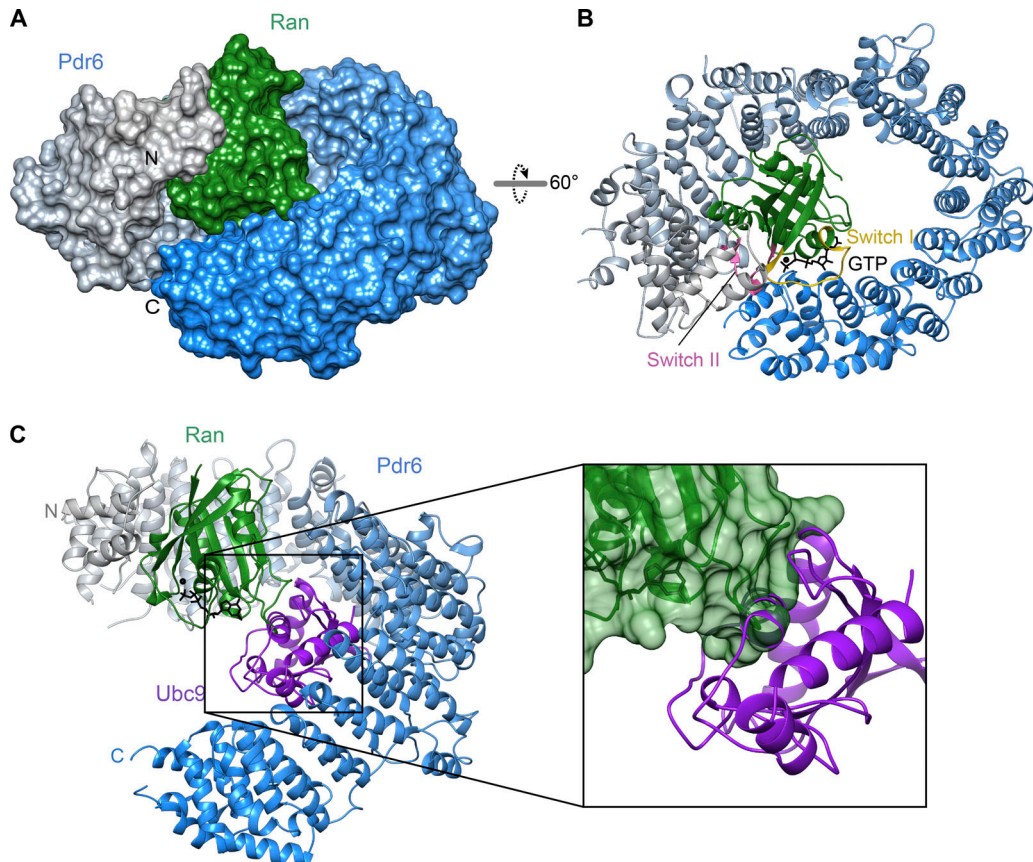

Figure 4. **Structure of RanGTP•Pdr6 complex. (A and B)** The complex is shown in the same orientation as in Fig. 2 (C and D) after alignment of Pdr6 with respect to the first three HEAT repeats. **(A)** The complex is shown as a surface representation. Pdr6 is depicted as in Fig. 2 while Ran is shown in green. **(B)** The complex is rendered as a ribbon representation. Switch I and II regions of Ran are shown as yellow and pink, respectively. GTP (black) is shown as sticks. **(C)** Pdr6•RanGTP and Pdr6•Ubc9 complexes were superposed via the N-terminal arch of Pdr6, and RanGTP was placed into the import complex. The resulting RanGTP•Pdr6•Ubc9 model is shown as a ribbon representation in the same orientation as in A. In the magnified image, Pdr6 was omitted for clarity, and a transparent surface is shown for RanGTP. Note the clash between Ubc9 and Ran, explaining why the two are antagonistic ligands.

complex (Fig. 5; Bono et al., 2010). Both, N- and C-terminal HEAT repeats contact switches I and II of Ran, thereby directly sensing Ran's nucleotide-bound state (Figs. 4 B, 5 C, and S2) and consequently coupling the RanGTPase cycle to a cargo-transport cycle. The N-terminal Ran-binding site (in HEATs 1, 2, and 3) is conserved in all so-far analyzed importin β family members (Fig. S3). The C-terminal one, however, reflects an adaptation to specific modes of cargo binding and release.

We have no direct knowledge of the cargo-release trajectory. However, it is tempting to assume that RanGTP initially binds to the N-terminal Ran-binding interface of Pdr6, which is accessible in the Pdr6•Ubc9 structure (Fig. 4 C), and thereby triggers the displacement of the import cargo. RanGTP-bound Pdr6 can then either exit the nucleus directly or recruit an export substrate such as eIF5A and then arrive as a ternary complex in the cytoplasm.

## Structure of the eIF5A•Pdr6•RanGTP export complex

The structures of the Pdr6 and Ran molecules within the ternary export complex are almost identical (RMSD, ~0.25 Å and 0.15 Å, respectively) to that of the Pdr6•RanGTP heterodimer. This suggests that eIF5A and Ran prefer the same conformation of Pdr6.

Both mammalian Xpo4 and yeast Pdr6 mediate nuclear export of eIF5A, both bind eIF5A from either yeast or human (Fig. 6 A), and, thus, both recognize conserved features of this cargo. It was, therefore, very surprising to see from our new structure that the two exporters recognize their common cargo in fundamentally different ways. eIF5A binds the two in different orientations, contacts them through different residues, and binds to opposite sides of the toroid (Fig. 6, C–E). Likewise, Pdr6 and Xpo4 respond very differently to mutations in eIF5A. While K50 and its hypusination are crucial for Xpo4 recognition (Lipowsky et al., 2000; Aksu et al., 2016), Pdr6 tolerates an unmodified or even a K50A-mutated residue (Fig. 6 B). Likewise, an H51A, R62E, or S75R exchange disrupts only the Xpo4 interaction but not the Pdr6 interaction. Conversely, we found the E42R exchange to be detrimental for Pdr6, binding but not for Xpo4 binding (Fig. 6 B).

Xpo4 recognizes eIF5A mainly by the intra-repeat loops of HEATs 11–16 while burying the positively charged regions of eIF5A, including the hypusine-containing loop (Fig. 6 D; Aksu et al., 2016). In the yeast eIF5A export complex, Pdr6 grasps eIF5A by N- and C-terminal arches, in particular, by the inter-repeat loops of HEATs 8–11 and the last helix of HEAT

**Figure 5. Comparison of RanGTP recognition by Pdr6 and Imp13. (A and B)** Ribbon representation of Pdr6•RanGTP (A) and Imp13•RanGTP complexes (B) are shown. Corresponding Pdr6 and Imp13 structures are aligned with respect to the first three HEAT repeats of the NTRs and shown in the same orientation as in Fig. 3. Pdr6 and Imp13 are shaded the same as in Fig. 3, while Ran and GTP are in green and black, respectively. **(C)** RanGTP sequence is shown in green, where Switch I and II regions are highlighted in yellow and pink, respectively. Salmon, magenta, and blue circles represent the Ran residues contacting Imp13, Xpo4, and Pdr6, respectively. Note that the Imp13•RanGTP structure contained yeast Ran (Gsp1p). For simplicity, corresponding human Ran residues are marked in this panel. Xpo4-contacting residues are determined from the RanGTP•Xpo4•eIF5A structure.

20 (Fig. 7, A and B; and Fig. S4). Pdr6 interacts with both domains of eIF5A, namely, with the N-terminal SH3-like domain and with the C-terminal oligonucleotide-binding (OB)–fold domain (Fig. S4). However, unlike Xpo4, Pdr6 does not shield the hypusine-containing loop. Instead, the hypusine-containing loop protrudes outward from Pdr6 (Fig. 7, A and B). Although Xpo4 and Pdr6 contact different residues of eIF5A (Fig. 6 E), Pdr6 also masks eIF5A's 25S RNA- and tRNA-binding interface (PDB 5GAK; Schmidt et al., 2016) and, therefore, can act as an eIF5A chaperone or inhibitor in the nucleus.

The N-terminal SH3-like domain of eIF5A constitutes the larger interaction interface and sits on a concave surface made up of the inter-repeat loops of HEATs 8–11 (Fig. S4 A). Most of these interactions involve hydrophobic contacts and are centered on W506$^{Pdr6}$ and K69$^{eIF5A}$. W506$^{Pdr6}$ docks into a pocket made up of V42$^{eIF5A}$, V60$^{eIF5A}$, I62$^{eIF5A}$, and K69$^{eIF5A}$ whereas K69$^{eIF5A}$ packs against W506$^{Pdr6}$, N511$^{Pdr6}$, M558$^{Pdr6}$, and W565$^{Pdr6}$. Additional polar contacts around these residues further stabilize this interaction.

RanGTP has to switch Pdr6 to a higher affinity for eIF5A in order to ensure that the cargo is performed of nuclei but does not reenter with the Ran-free receptor. Such a switch can occur by two (not mutually exclusive) principles. First, Ran promotes cargo binding to the exporter by contributing part of the cargo-binding interface. This applies to the export of importin α, tRNA, or pre-miRNAs (Matsuura and Stewart, 2004; Cook et al., 2009; Okada et al., 2009). Second, Ran stabilizes the exportin or bi-portin in a high-energy (spring-loaded) "nuclear conformation" with a then-active cargo-binding site. This applies to CRM1 and Xpo4 (Monecke et al., 2009; Aksu et al., 2016).

In our case, it appears that the recognition of the C-terminal OB-fold domain of eIF5A is the key to this Ran control. The OB-fold domain is clamped between the N- and C-terminal arches of Pdr6 and positioned directly adjacent to Ran. Pdr6-Ran forms a positively charged surface that is complementary to negatively charged residues of eIF5A at this interface (Fig. 7 C). In particular, D94$^{eIF5A}$ and D96$^{eIF5A}$ contact R1057$^{Pdr6}$ while E119$^{eIF5A}$, D122$^{eIF5A}$, and E130$^{eIF5A}$ approach K99$^{Ran}$, R132$^{Ran}$, and R134$^{Ran}$ although at a distance that might include additional water

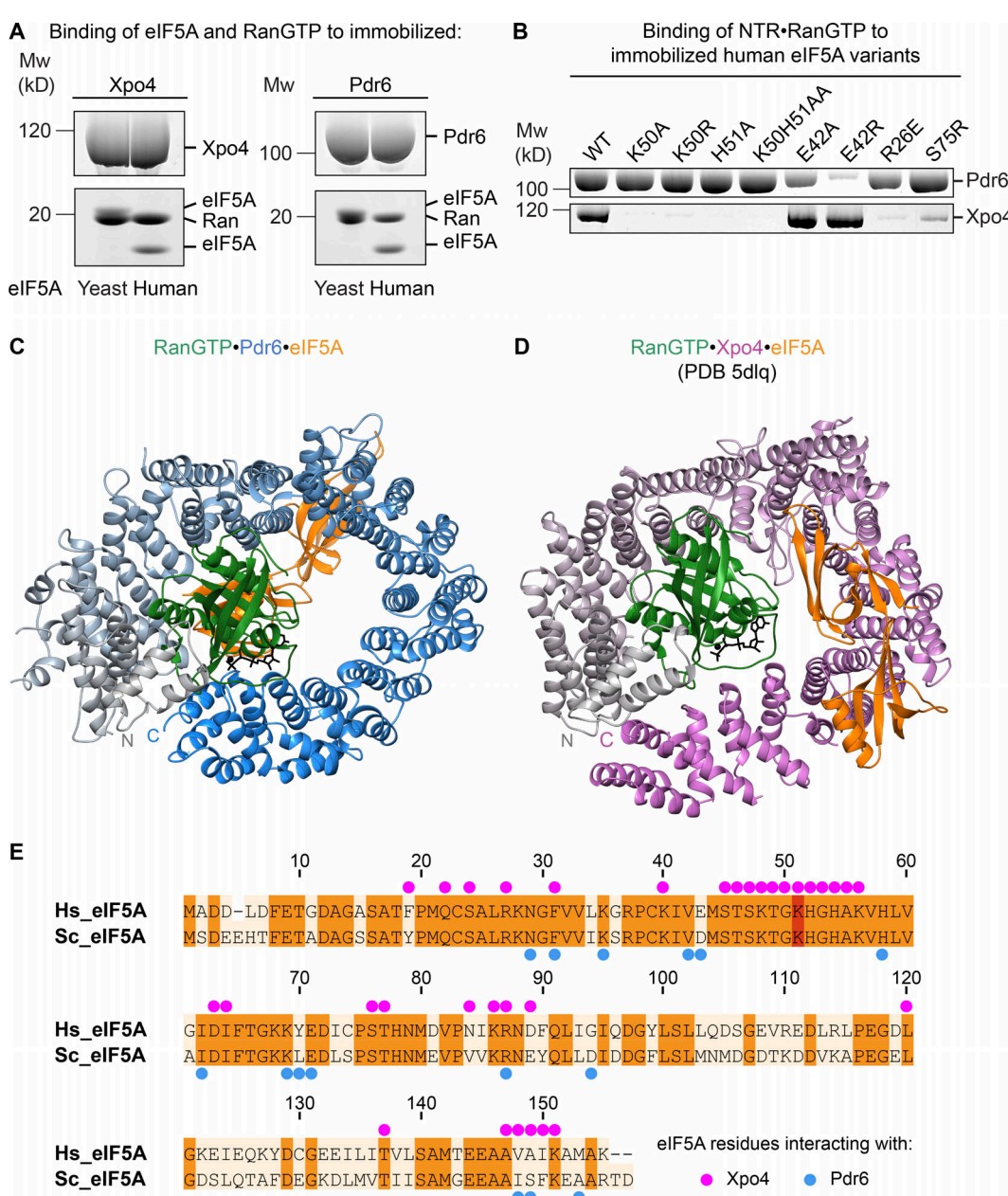

**Figure 6. Comparison of eIF5A recognition by Pdr6 and Xpo4. (A)** H_14-ZZ-bdSUMO–tagged Pdr6 and Xpo4 (1 µM) were incubated with either human or yeast eIF5A (2 µM) in the presence of RanGTP (2 µM). Formed complexes were retrieved via tagged NTRs, eluted by (the tag-cleaving) bdSENP1 protease, and analyzed by SDS-PAGE and Coomassie blue staining. The identity of each protein band is as indicated in the figure. Note that yeast eIF5A and Ran bands partially overlap in both panels. **(B)** 1 µM NTR and RanGTP was incubated with 0.75 µM ZZ-bdSUMO–tagged eIF5A wild type or mutants in a 50 mM NaCl buffer. Formed complexes were retrieved via tagged NTRs, eluted by bdSENP1 protease, and analyzed by SDS-PAGE and Coomassie blue staining. **(C and D)** Ribbon representation of RanGTP•Pdr6•eIF5A (C) and RanGTP•Xpo4•eIF5A complexes (D) are shown. Structures are aligned with respect to the first three HEAT repeats of Pdr6 and Xpo4 and shown in the same orientation as in Fig. 2 D. Pdr6 is depicted with a color gradient from gray (N-terminus) to blue (C-terminus) while Xpo4 is from gray to magenta. Ran and eIF5A are shown in green and orange, respectively. **(E)** Sequence alignment of human and yeast eIF5A. Identical residues are shaded in orange and the hypusine-modified lysine in red. Blue and magenta dots represent eIF5A residues that contact Pdr6 and Xpo4, respectively.

molecules. Furthermore, I148^eIF5A and S149^eIF5A contact N441^Pdr6 and stabilize the OB-fold interactions (Fig. S4 B).

Ran does not only interact with eIF5A directly but also promotes Pdr6-eIF5A contacts by bringing the N- and C-terminal arches of Pdr6 (which stay apart in the Pdr6 • Ubc9 complex and probably also in Pdr6 apo structure) closer to each other. Therefore, the cooperativity is accomplished by the combination of an allosteric mechanism and free energy supplied by the cargo–RanGTP interaction.

## Discussion

Pdr6 has so far been a rather poorly studied shuttling NTR with just two import cargoes, subunits of TFIIA and a Wtm1

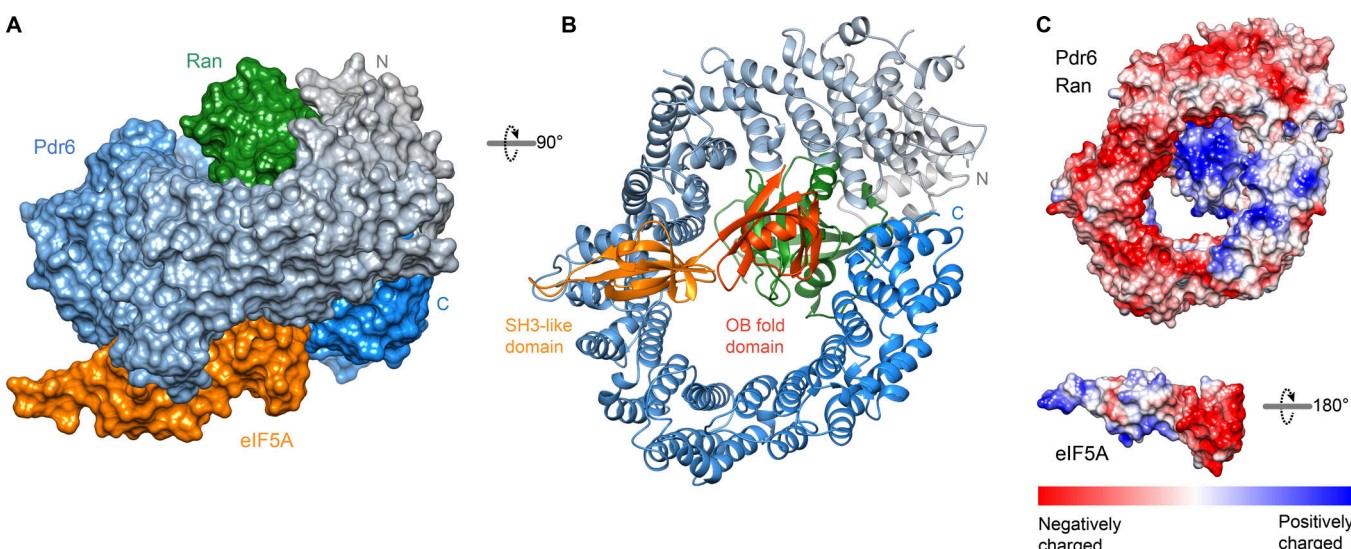

**Figure 7. Structure of the yeast eIF5A export complex. (A and B)** View of the trimeric export complex in two different orientations. **(A)** The complex is shown as a surface representation. Pdr6 is depicted with a color gradient from gray (N-terminus) to blue (C-terminus) while Ran is shown in green, and eIF5A is in orange. **(B)** The complex is rendered as a ribbon representation. Domains of eIF5A are labeled and shaded accordingly. **(C)** Pdr6•RanGTP and eIF5A are shaded according to electrostatic potential with a color gradient from red (negatively charged, −10 kcal/mol) to blue (positively charged, +10 kcal/mol). Pdr6•RanGTP is shown in the same orientation as in B while eIF5A is rotated 180°.

ribonucleotide reductase complex, being known. In the accompanying paper (Vera Rodriguez et al., 2019), we reevaluated its cargo spectrum and identified not only Ubc9 as a novel import cargo but also export cargoes, namely, the translation elongation factors eIF5A and eEF2, as well as the membrane trafficking components Pil1 and Lsp1. This defines Pdr6 as a bidirectional NTR or a biportin. In this study, we report the structures of key transport intermediates, namely, of the Ubc9•Pdr6 import complex, of the Pdr6•RanGTP intermediate, and of the eIF5A•Pdr6•RanGTP export complex. These structures not only illuminate how the RanGTPase system drives Pdr6-dependent transport in and out of the cell nucleus but also reveal rather unusual modes of cargo recognition.

The transport signals recognized by Pdr6 are folded domains. This puts Pdr6 apart from Importin α and CRM1, which recognize linear peptides, namely, canonical NLSs and NESs, respectively. It can be easily understood how Importin α and CRM1 can each handle a broad spectrum of cargoes if these carry the same type of NLS or NES for docking into the same dedicated binding sites of the transporters.

It is, however, rather unclear, how Pdr6 can carry multiple import and export cargoes. Since these have different folds and share no structural similarity, we would assume that different binding sites are used for different cargoes, whereby each of these sites should respond to RanGTP. Such multispecificity is a fascinating case of molecular recognition. However, its comprehension requires solving more structures, namely, of the export complexes with eEF2 and the BAR domain proteins Lsp1 and Pil1, as well as import complexes with TFIIA and Wtm1.

We focused on two transport cargoes, namely, eIF5A and Ubc9. eIF5A is a rather small (17-kD) protein that functions in cytoplasmic translation but readily leaks into nuclei. Pdr6-mediated export should be seen as part of the cell's effort to suppress nuclear translation and not to waste cellular resources in mislocalized proteins. In addition, it is well possible that the inappropriate presence of eIF5A inside nuclei interferes with nuclear processes related, for example, to RNA metabolism and/or ribosome biogenesis. Pdr6 exports eIF5A with its 25S RNA- and tRNA-binding sites being blocked. This poses the question of why it has to be exported in such a chaperoned fashion and not through an appended NES as hundreds of other cytoplasmic proteins.

We see several possible answers. Translation is a highly optimized process with highly optimized translation factors. Possibly, an NES appendix compromises eIF5A's activity in translation. Considering that translation is one of the most resource-requiring cellular processes, even a small negative effect could have a negative impact on fitness. An NTR that adopts to a fully optimized translation factor is, therefore, a good solution to such a problem. Furthermore, chaperoning the cargo might help to cleanly "extract" eIF5A from the nucleus without coexporting interacting molecules.

Our structure of the Ubc9•Pdr6 import complex revealed that Pdr6 is transporting Ubc9 in an inhibited state with blocked interaction sites for the E1 SUMO-activating enzyme with SUMO E3 ligases and SUMOylation substrates. This raises the questions of why this should be an advantage and why Ubc9 import does not occur along the classical importin α/β pathway with a canonical NLS being appended to the molecule.

We see two possible answers. First, the addition of a (typically lysine-rich and positively charged) NLS might compromise the SUMOylation specificity of Ubc9, for example, through inappropriate electrostatic interactions or by acting as an intramolecular SUMO acceptor. Second, transporting Ubc9 through an appended NLS would imply that Ubc9 remains enzymatically active while bound to the import receptor. This might be

dangerous because the shuttling transport receptor might then become a target of inappropriate (over-) SUMOylation. The same would apply to NPC components that contact the transiting NTR•Ubc9 complex, whereby a SUMOylation of FG repeat domains could compromise the selectivity of the FG-based permeability barrier of NPCs. Diverting Ubc9 import from the major (Importin α/β-dependent) pathway and transporting it in a safe-locked state appears, therefore, a perfect solution to this problem. This assumption is supported by the fact that, with Imp13, mammals found a fully independent solution for Ubc9 import, which features a fundamentally different mode of cargo recognition yet follows the same principle of carrying the SUMO-conjugating enzyme safely locked in a state of inactivity.

## Materials and methods

### Protein expression and purification

Recombinant mouse Xpo4, human RanQ69L (5–180), human eIF5A, and *S. cerevisiae* eIF5A variants, as well as human Imp13, were expressed and purified as previously described (Aksu et al., 2016; Vera Rodriguez et al., 2019).

For crystallization, Pdr6 was expressed as an N-terminal His14-ZZ-bdSUMO fusion in *Escherichia coli* Top10 F′. The cells were resuspended in buffer A (50 mM Tris/HCl, pH 7.7, 500 mM NaCl, and 2 mM DTT) and lysed by sonication on ice. The clarified lysate was incubated with a Ni (II) chelate matrix in the presence of 15 mM imidazole/HCl, pH 7.7. The matrix was loaded to a gravity flow column. The column was first washed with buffer A supplemented with 25 mM imidazole, and Pdr6 was later eluted by 250 nM of the tag-cleaving bdSENP1 protease (Frey and Görlich, 2014). The eluate was subjected to a Superdex 200 16/60 gel filtration column (GE Healthcare) equilibrated with buffer B (20 mM Tris/HCl, pH 7.7, 150 mM NaCl, and 2 mM DTT). For phasing, selenomethionine-substituted Pdr6 was expressed in BL21 cells grown in minimal medium supplemented with lysine, phenylalanine, threonine, isoleucine, leucine, valine, and selenomethionine. Selenomethionine-labeled Pdr6 was purified using the same protocol as for unmodified Pdr6. The DTT concentration was increased to 5 mM during the purification.

For binding assays, Pdr6, mouse Xpo4, and human Imp13 (see Table S1 for constructs) were expressed in *E. coli* Top10 F′ cells and purified by following the above protocol. After washing the columns, the NTRs were eluted either by bdSENP1 protease or by imidazole elution using buffer A supplemented with 500 mM imidazole.

Ubc9 and eIF5A variants were expressed in NEB Express cells and purified by Ni (II) chelate chromatography and elution by a tag-cleaving protease (for binding assays) or imidazole elution (for crystallization).

RanQ69L (residues 5–180) was expressed as an N-terminal His14-ZZ-bdSUMO fusion in *E. coli* Top10 F′ cells. The cells were lysed in buffer C (50 mM Hepes/KOH, pH 8.2, 500 mM NaCl, 2 mM MgCl$_2$, and 2 mM DTT). Ran was bound in the presence of 20 mM imidazole from the cleared lysate to a Ni (II) chelate matrix and eluted with bdSENP1 protease as described above.

### Reconstitution and crystallization of complexes

To prepare the Pdr6•Ubc9 complex, His14-bdNEDD8–tagged Ubc9 was mixed with a 1.1-molar excess of Pdr6 in 20 mM Tris/HCl, pH 7.7, 50 mM NaCl, and 5 mM DTT. After overnight incubation on ice, the sample was supplemented with 5 mM imidazole, and the complex was immobilized on a Ni (II) chelate matrix via tagged Ubc9. Unbound proteins were removed; Ubc9 and the bound Pdr6 were eluted by bdNEDP1 protease. The eluate was subjected to a Superdex 200 16/60 gel filtration column equilibrated with 20 mM Tris/HCl, pH 7.7, 50 mM NaCl, and 2 mM DTT. The purified complex was concentrated to 10 mg/ml.

Diffraction-quality crystals were obtained at 18°C by mixing 1 µl protein solution with 1 µl reservoir solution containing 50 mM sodium cacodylate, pH 5.7, 20 mM MgCl$_2$, and 3–5% ethanol. Crystals were slowly transferred to a cryoprotectant solution (50 mM sodium cacodylate, pH 5.70, 20 mM MgCl$_2$, 15% ethanol, and 30% polyethylene glycol [PEG] 400) and flash-frozen in liquid nitrogen.

For Pdr6•RanGTP•eIF5A complex formation, His14-ZZ-bdNEDD8–tagged eIF5A was mixed with 1.2-molar excess of Pdr6 and RanGTP in 20 mM Tris/HCl, pH 7.7, 50 mM NaCl, 5 mM MgCl$_2$, and 5 mM DTT. The complex was obtained as described above. After size exclusion chromatography in 20 mM Tris/HCl, pH 7.7, 50 mM NaCl, 5 mM MgCl$_2$, and 2 mM DTT, the purified complex was concentrated to 12 mg/ml and supplemented with 2.5 mM Tris(2-carboxyethyl)phosphine hydrochloride.

Initial Pdr6•RanGTP crystals were obtained at 18°C by mixing 1 µl Pdr6•RanGTP•eIF5A complex with 1 µl reservoir solution containing 50 mM sodium acetate, pH 5.5, 220 mM MgCl$_2$, 19% pentaerythritol propoxylate (17/8 PO/OH), and 10% 2-Methyl-2,4-pentanediol. These crystals lacked eIF5A, which was probably dissociated from the rest of the complex during crystallization due to high salt concentration. Similar crystals were also obtained by using 10 mg/ml Pdr6•RanGTP complex prepared as described above but omitting eIF5A. These crystals were directly flash-frozen in liquid nitrogen.

Pdr6•RanGTP•eIF5A crystals were obtained at 18°C by microseeding hanging drops by mixing 1 µl of the protein solution with 1 µl reservoir solution containing 100 mM sodium acetate, pH 5.3, and 30% PEG 300. Crystals were slowly transferred to a cryoprotectant solution (100 mM sodium acetate, pH 5.3, 45% PEG 300, and 10% glycerol) and flash-frozen in liquid nitrogen.

### Structure determination and analysis

All diffraction data were collected at beamline X10SA at the Swiss Light Source. For the Pdr6•RanGTP structure, seven datasets from five crystals were merged, integrated, and scaled with XDS (Kabsch, 2010). SHELXD (Schneider and Sheldrick, 2002) was used to locate the 26 selenium sites. Initial phases were obtained by MR with PHASER (McCoy, 2007) using Ran (PDB 3GJX; Monecke et al., 2009) as the search model. The resulting information and position of selenium atoms were used to obtain the electron density map in AutoSol Wizard (Terwilliger et al., 2008, 2009) in the Phenix suite (Adams et al., 2002). Model building was performed with RESOLVE and BUCCANEER (Cowtan, 2006) by using AutoBuild Wizard (Terwilliger et al.,

2008) in Phenix and with COOT (Emsley and Cowtan, 2004). Iterative cycles of refinement using phenix.refine (Afonine et al., 2012) were done after each round of model building.

The Pdr6•Ubc9 dataset was processed in XDS (Kabsch, 2010). The structure was solved by MR in PHASER (McCoy, 2007). For a successful MR solution, Pdr6 structure was separated into two rigid bodies (residues 3–510 and 560–1,076). Placement of Pdr6 revealed the electron density for Ubc9. The structure of yeast Ubc9 (PDB 2GJD; van Waardenburg et al., 2006) was placed into electron density manually. The model was manually adjusted in COOT (Emsley and Cowtan, 2004), rigid-body refined, and then subjected to all-atom refinement for five cycles in phenix.refine (Afonine et al., 2012).

The RanGTP•Pdr6•eIF5A dataset was processed in XDS (Kabsch, 2010). Analysis of the dataset in Xtriage revealed 28% twinning. The structure was solved by MR in PHASER (McCoy, 2007) using the Pdr6•RanGTP structure described above. Density modification in AutoBuild (Terwilliger et al., 2008) revealed the electron density for eIF5A. A yeast eIF5A homology model (created in SWISS-MODEL; Bordoli et al., 2009) based on human eIF5A (PDB 5DLQ; Aksu et al., 2016) was manually placed into the density. The model was manually adjusted in COOT (Emsley and Cowtan, 2004) and all-atom refined for 20 cycles using h,-h,-k,-l twin law in phenix.refine (Afonine et al., 2012). The quality of all final models was assessed with MolProbity (Chen et al., 2010). All figures were prepared using UCSF Chimera (Pettersen et al., 2004).

### Binding assays

For Ubc9 compatibility assays (Fig. 3 D), 1 µM $H_{14}$-ZZ-bdSUMO–tagged NTR (Pdr6 or Imp13) was incubated with either human or yeast Ubc9 (2 µM).

For eIF5A compatibility assays (Fig. 6 A), 1 µM $H_{14}$-ZZ-bdSUMO–tagged NTR (Pdr6 or Xpo4) was incubated with either human or yeast eIF5A (2 µM) in the presence of 2 µM RanGTP.

For eIF5A mutant tests (Fig. 6 B), 1 µM NTR (Pdr6 or Xpo4) and RanGTP were incubated with 0.75 µM ZZ-bdSUMO–tagged eIF5A wild type or mutants in a 20 mM Tris/HCl, pH 7.7, 100 mM NaCl, 2 mM MgOAc, and 2 mM DTT buffer.

Formed complexes were retrieved via tagged proteins by anti–Protein A beads, eluted by (the tag-cleaving) bdSENP1 protease, and analyzed by SDS-PAGE and Coomassie blue staining.

### Data depositions

The coordinates and structure factors have been deposited in the PDB with accession code 6Q82 for Pdr6•RanGTP, 6Q83 for Pdr6•Ubc9, and 6Q84 for RanGTP•Pdr6•eIF5A.

### Online supplemental material

Fig. S1 shows Pdr6•Ubc9–binding interfaces and demonstrates the potentially interacting residues. Fig. S2 depicts recognition of Ran's switches by Pdr6 N- and C-terminal residues. Fig. S3 highlights RanGTP-interacting regions of NTRs and shows a sequence alignment of the first three HEAT repeats. Fig. S4 shows Pdr6-eIF5A–interacting residues. Table S1 lists the plasmids used in this study.

## Acknowledgments

We thank Gaby Hawlitschek and Jürgen Schünemann for excellent technical help, the crystallization facility of the Max Planck Institute for Biophysical Chemistry for splendid support, the staff of synchrotron beamlines at the Swiss Light Source (Villigen, X10SA, PXII) for assistance during data collection, and Trevor Huyton for data collection of the RanGTP•Pdr6•eIF5A crystal.

This work was supported by the Max-Planck-Gesellschaft and the Deutsche Forschungsgemeinschaft (SFB 860/B03).

The authors declare no competing financial interests.

Author contributions: M. Aksu, S. Trakhanov, A. Vera Rodriguez, and D. Görlich designed and performed experiments and analyzed and interpreted data; M. Aksu and D. Görlich wrote the paper; A. Vera Rodriguez contributed to writing; and all authors approved the manuscript.

Submitted: 18 December 2018

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
