## [Reviewer comments · The Journal of Cell Biology]

Structural basis for the nuclear import and export functions of the biportin Pdr6/Kap122

Metin Aksu, Sergei Trakhanov, Arturo Vera-Rodriguez, and Dirk Görlich

Corresponding Author(s): Dirk Görlich, Max Planck Institute for Biophysical Chemistry

Review Timeline:

Submission Date:	2018-12-18
Editorial Decision:	2019-02-07
Revision Received:	2019-03-26

Monitoring Editor: Larry Gerace

Scientific Editor: Melina Casadio

Transaction Report:

DOI: 10.1083/jcb.201812093

February 7, 2019

RE: JCB Manuscript #201812093

Prof. Dirk Görlich
Max Planck Institute for Biophysical Chemistry
Am Faßberg 11
Göttingen 37077
Germany

Dear Prof. Görlich,

Thank you for submitting your manuscript entitled "Structural basis for the nuclear import and export functions of the bioprotein Pdr6/Kap122". Thank you very much for your patience with the peer review process -- we are sorry for the significant delay in communicating our decision to you. The manuscript was evaluated by a mixed panel of nuclear pore biology and nuclear transport experts, either from the structural or cell biological standpoint, and their comments are appended below. Reviewers #1-2-3 also reviewed your co-submission entitled "Bidirectional nuclear transport by Pdr6 discovered through an engineered SUMO/protease system".

As you will see, there was an enthusiastic response to your submissions, both to the scientific advances presented and the quality of the data. The reviewers indicated that your work on the structure of Prd6 in complex with an import cargo and export cargo/Ran substantially broadens the current structural understanding of NTRs. They asked a number of interesting and important questions that seem addressable by figure edits and clarifications in the text.

We would be happy to publish your paper in JCB pending changes to address the reviewers' comments and pending final revisions necessary to meet our formatting guidelines (see details below).

1) eTOC summary: please provide a 40-word summary that describes the context and significance of the findings for a general readership should be included on the title page. The statement should be written in the present tense and refer to the work in the third person.

Please let us know if suggestions would be helpful.

2) Figure formatting:

Molecular weight or nucleic acid size markers must be included on all gel electrophoresis. Please add molecular weight with unit labels on the following panels: 3D, 6AB

3) Materials and methods: Should be comprehensive and not simply reference a previous publication for details on how an experiment was performed. Please provide full descriptions in the text for readers who may not have access to referenced manuscripts.

- Please provide all details for the protein purification and expression studies as well as the Structure determination and analysis methods, even if described in other published work.

4) A summary paragraph of all supplemental material should appear at the end of the Materials and methods section.

5) Conflict of interest statement: JCB requires inclusion of a statement in the acknowledgements regarding competing financial interests. If no competing financial interests exist, please include the following statement: "The authors declare no competing financial interests." If competing interests are declared, please follow your statement of these competing interests with the following statement: "The authors declare no further competing financial interests."

6) Author contributions: A separate author contribution section is required following the Acknowledgments in all research manuscripts. All authors should be mentioned and designated by their full names. We encourage use of the CRediT nomenclature.

A. MANUSCRIPT ORGANIZATION AND FORMATTING:

Full guidelines are available on our Instructions for Authors page, <http://jcb.rupress.org/submission-guidelines#revised>. **Submission of a paper that does not conform to JCB guidelines will delay the acceptance of your manuscript.**

B. FINAL FILES:

-- High-resolution figure and video files: See our detailed guidelines for preparing your production-ready images, <http://jcb.rupress.org/fig-vid-guidelines>.

Thank you for your attention to these final processing requirements. Please revise and format the manuscript and upload materials within 7 days. (An extension to resubmit both papers together would be fine with us.) Please include a point-by-point response to all the reviewers' comments with your resubmission.

Thank you for this interesting contribution, we look forward to publishing your paper in the Journal of Cell Biology.

Sincerely,

Larry Gerace, PhD
Monitoring Editor, Journal of Cell Biology

Melina Casadio, PhD
Senior Scientific Editor, Journal of Cell Biology

Reviewer #1 (Comments to the Authors (Required)):

Aksu et al. describe the structural basis of nucleocytoplasmic transport mediated by the *S. cerevisiae* karyopherin Pdr6. In the accompanying manuscript, the authors identified the SUMO-conjugating enzyme UBC9 and the translation factor eIF5A as nuclear import and export cargo of Pdr6, respectively. To provide detailed insight into the mechanism of cargo binding and release, the authors determined the crystal structures of the Pdr6•RanGTP, Pdr6•UBC9, and the Pdr6•RanGTP•eIF5A complexes to a resolution of 2.9 Å, 4.53 Å, and 3.7 Å, respectively. Whereas the Pdr6•RanGTP structure closely resembles the binding mode of its mammalian homolog Imp13•RanGTP, the complexes of Pdr6•UBC9 and Pdr6•RanGTP•eIF5A show striking differences to their mammalian counterparts. Both UBC9 and eIF5A bind Pdr6 at entirely different interaction interfaces compared to the mammalian Imp13•UBC9 and Xpo4•RanGTP•eIF5A complexes. Interestingly, Aksu et al. demonstrate biochemically that *S. cerevisiae* and human eIF5A can form chimeric complexes with *S. cerevisiae* Pdr6 and its mammalian homolog Xpo4. In line with their structural studies, a mutational analysis shows that eIF5A indeed binds to different regions on Pdr6 and Xpo4. The authors then conclude that nuclear import of UBC9 and nuclear exclusion of eIF5A evolved more than once and independent from each other. Overall, the presented manuscript is a well conducted interdisciplinary study, with a very high technical quality as expected from the Gorlich group. The determined crystal structures constitute an important advance in the field. However, to make the manuscript more accessible to the broad readership of JCB, the authors should address the following specific points:

1. In the abstract the authors state: "This suggests that non-canonical import of UBC9 and active nuclear exclusion [...]" What is the term "non-canonical" based on? To date, the only (published) available insight in the UBC9-karyopherin interaction is based on the UBC9•Imp13 structure. It would be beneficial to clarify the term given that, to the reviewer's knowledge, there is only one counter-example of UBC9 import in the literature.
2. The authors claim that "Pdr6 bound human Ran and *S. cerevisiae* Gsp1p equally well [...]" Do the authors show this anywhere? What are the dissociation constants? Either quantitative data should be included, or the text should be revised accordingly.
3. "Pdr6 is an all alpha-helical protein built of 19 canonical HEAT repeats followed by three helices (termed HEAT 20) at its very C-terminus (Fig. 2A-D)." The C-terminus of a protein is the last amino acid, terminated by a carboxyl group. The authors should refer in their description maybe to "its very C-terminal region".

4. "Our Pdr6•RanGTP structure is indeed incompatible with Ubc9-binding (Fig.4)." Whereas the data are unambiguous, the visual representation in Figure 4C is not particularly meaningful. It would be better to highlight the clash with a magnified view to make the incompatibility of Ubc9 and RanGTP more accessible to the reader.
5. "As discussed below, we assume that Ran puts the superhelical structure under tension, which could explain why the binary RanGTP Pdr6 interaction is only of moderate affinity (~230 nM [...])." It would be informative to compare this with the affinities of other RanGTP-NTR interactions.
6. "The N-terminal Ran-binding site is conserved in all so far analyzed importin beta family members. The C-terminal one, however, reflects an adaptation to specific binding modes of cargo binding and release." The authors should include a figure panel to make this conclusion more accessible to the reader.
7. "The two recognize conserved features on this cargo - to the extent that Pdr6 binds yeast and human eIF5A equally well (Fig. 6A)." Figure 6A shows that Pdr6 binds human and yeast eIF5A, but does not allow for any quantitative statements. The current phrasing suggests that human and yeast eIF5A bind with similar affinities, which has not been tested. The authors should either revise their statement or provide quantitative data.
8. "First, Ran forms a common interaction interface with the cargo and the released binding energy promotes recruitment of the other NTR-ligand [...]." This sentence is confusing and requires clarification.
9. Details for the experiments shown in Figure 6A and 6B are missing in the method section and should be added.
10. Figure 2E and 7C depict electrostatic surface potentials. The extreme points of the scale need to be indicated, or at least mentioned in the figure legend.
11. Figure S1 visualizes the interaction of Pdr6 and Ubc9 with a magnified view in panel A that highlights salt bridges. In the absence of showing electron density in a separate figure panel, the accuracy of the depicted salt bridges is questionable at a resolution of 4.53 Å. At this somewhat low resolution I would not expect such a salt bridge to be well resolved.
12. The yellow residue labels in Figure S2 are basically unreadable and should be revised.
13. Crystallographic table, data and refinement statistics and structure quality. Unit cell dimensions and R factors should be rounded to one decimal and B factors should be rounded to full integers. Rmerge values and Rpim values (currently missing) should be included for all datasets and reported in percentages. The high-resolution cutoffs should be determined with the I/σ in the highest resolution shell. While modern Pilatus detectors allow the usage of all data, an I/σ in the highest resolution shell of <1 is actually less data than background (1). Thus, the high-resolution cutoff should be adjusted to have an I/σ in the highest resolution shell of > 1 . CC1/2 values (currently missing) should be included for all datasets as well. One of the datasets has an unusual multiplicity of ~400. Is this really correct? Ramachandran outliers should be 0%, especially for low resolution crystal structures as in this case. MolProbity scores should be reported for all structures.
14. English and grammar. The manuscript would benefit from substantial editing by the senior

author.

Reviewer #2 (Comments to the Authors (Required)):

The two manuscripts by Asku et al and Vera-Rodriguez et al define yeast Prd6 as a bidirectional NTR and provide the structural basis for its cargo recognition. Each of the two manuscripts is experimentally very extensive and taken together they define novel yeast export pathway that was previously unknown.

Major comments on Vera-Rodriguez et al:

The authors have developed an elegant readout for the specificity of mutagenized SUMO proteases towards mutagenized SUMO tags and used it to evolve SUMO tag / SUMO protease pairs that resist cleavage of endogenous SUMO proteases in human and yeast cells. The authors demonstrate that their system is superior to existing sumo affinity tags, in terms of cleavage kinetics and stability, even in human cells. The breadth of this screen is impressive and I believe it has good chances to make a high impact, not only because of its immediate outcome but also because it might be adapted to other protease substrate pairs.

The authors employ their system to identify cargos of yeast Prd6. They validate ubc9 as an import, as well as Phil1, Lsp1, eIF5A, eEF2 as export cargos. The experiments presented are of very high quality and convincing. The concept of bidirectional active nuclear transport is per se not novel, and the authors cite the respective papers on Imp13, etc. in the introduction. This capacity however is newly identified for Prd6 by this study, which is a very exciting finding that broadens our understanding of the nuclear transport machinery.

I most enthusiastically recommend this paper for publication in JCB after minor revisions.

Minor comments on Vera-Rodriguez et al:

- Fig. 1A: This would be more straight forward to comprehend if the cartoon would include two arrows pointing to reaction products and their activity in case PCSfor or PCSagainst are cleaved. You might also indicate the degron that uses the N-end rule in the cartoon.
- Along similar lines, a flow chart illustrating the conceptual layout of the screen would be helpful. The authors have included very few supplementary items thus far and still have space.
- It took me ages to figure out that the numbers shown on top of the lanes in Figure 3A correspond to ones shown in Figure 3B in tiny fonts size. Please consider making this more user friendly, e.g. mention in (A) that this is explained in (B), or better swap both panels.
- Regarding the term 'biportin', I am not sure if suggesting yet another term is a good approach for reducing the confusion about nomenclature in this field. What is a transportin than? Would one have to rename Imp13, Xpo4, Xpo7 and Msn5? But have it your way ...
- Discussion: A model in which expelling translation factors from the nucleus is important to prevent nuclear translation generously assumes that pre-ribosomes (when given a chance) would have translational activity. But is this really the case? Even if so, expelling initiation factors would be sufficient to prevent nuclear translation and it is not clear why there should be evolutionary pressure on all translation factors. In that sense, the authors explanation of 'metabolic burden of mis-localized proteins' or maybe better 'kinetic burden' of reduced concentrations of translation factors in the cytosol, is a simpler and in my view also more plausible interpretation.

Major comments on Asku et al:

The authors have solved structures of key intermediates of the Prd6 transport cycle, specifically of ubc9-Prd6 import complex, RanGTP-Prd6 without cargo and RanGTP-Prd6-eIF5A export complex. The structures show some differences to their metazoan counterparts, that is the respective Xpo4 and Imp13 structures. These structures reveal the molecular basis for both Prd6-dependent pathways, as defined in the accompanying paper, in very high detail. The unconventional, bidirectional transport scheme of these pathways make this a very interesting paper, that does considerably go beyond our present structural knowledge of NTRs and their substrate binding mode. The fact that Prd6 also covers key interfaces of its substrates suggesting a chaperoning function is highly interesting! On a general note, solving as many cargo-NTR complexes as possible is important for this field. Since there is such a huge number of them, the existing structural repertoire is still quite limited.

I most enthusiastically recommend this paper for publication in JCB after minor revisions.

Minor comments on Asku et al:

- Page 6: Please make clear in the main text if the 'complex of full-length Pdr6 and the GTPase-deficient RanQ69L' contains yeast or human Ran. Does not matter much since its super-conserved, but should be transparent. Likewise, please clarify for Prd6-RanGTP complex.

- Page 7: Regarding the statement 'Furthermore, we found that Pdr6 imports UBC9, exports eIF5A and thus combines Imp13 and Xpo4 functions. This could suggest that Imp13 and Xpo4 evolved by gene duplication and diversification from an ancestral Pdr6 gene.' If you first say 'gene duplication and diversification', you make one assume that the ubc9 and eIF5A binding function existed before gene duplication, which would be inconsistent with that function having 'appeared twice in evolution'. What you mean to say is that most likely an ancient NTR gene was duplicated and the resulting genes adopted ubc9 and eIF5A binding functions independently, correct? Please consider making this clearer.

On another note, the simplest imaginable evolutionary trajectory is not necessarily the most likely. In principle it could be both, convergent or divergent evolution, right?

Comment on both papers:

Why would Phil1, Lsp1, eIF5A and eEF2 be exported by a different pathway than crm1-dependent export?! The authors suggest that an NES might interfere with the activity of a 'highly optimized translation factor', however, this argument does not apply to all Prd6 cargos. Could this have to do with the proposed nuclear chaperoning function of i.e. the hypusine-containing interface? But what is the evidence really that exposing this interface in the nucleus is harmful? Can one experimentally test this idea? Would exposing this interface in the nucleus by overexpressing eIF5A fused to a classical NLS (on top of the endogenous protein) cause a reduced fitness phenotype that depends on the expression level? Likewise, several other hypotheses spelled out in the discussion of mainly Asku et al, could well be tested. In more general terms, the strength of both papers is that they provide a very solid base to test the physiological relevance of the Prd6-dependent pathways. The two papers however do not yet capitalize on this capacity, which would be quite interesting because of the relevance for metazoan Xpo4 and Imp13. Both papers however already comprise an experimental tour de force. Extending the scope even further is certainly not required for publication.

Reviewer #3 (Comments to the Authors (Required)):

This manuscript describes crystal structure analysis of yeast UBC9/Pdr6 import complex,

RanGTP/Pdr6 heterodimer and RanGTP/Pdr6/eIF5A export complex. UBC9 and eIF5A are the new cargoes that was identified by the same authors' group in the accompanying paper. These yeast structures were compared with crystal structures of mammalian UBC9/Imp13 import complex and RanGTP/eIF5A/Exp4 export complex that was previously solved. Recognition of UBC9 by yeast Pdr6 differs from that of UBC9 by mammalian Imp13, and yeast eIF5 by Pdr6 from that of mammalian eIF5 by Exp4. The authors discuss evolutionary aspects on cargo recognition by nuclear transport receptors (NTRs).

Regarding cargo recognition, how the diverse cargoes can be recognized by the same NTR is an intriguing problem. At the same time, evolutionary aspects of cargo recognition by NTR is also important untouched subject. The manuscript is well written and data are clear. I support aim of this study.

I have one question,
Fig6A, why yeast eIF5A do not bind to Pdr6 and Xpo4?

Max-Planck-Institut für Biophysikalische Chemie

Dr. Dirk Görlich · Managing Director

Am Fassberg 11 · D-37077 Göttingen · E-mail: goerlich@mpibpc.mpg.de · Tel.: ++49 551 2012401

To the Editors of The Journal of Cell Biology
- via the internet -

Göttingen, 26th of March 2019

JCB manuscripts #201812091 and #201812093

Dear Editor, dear Reviewers,

Thank you very much for the very positive evaluation of our two manuscripts and the valuable input for further improvements. We have prepared accordingly revised versions, which we hope can now be accepted for publication. A list of changes and answers to all points raised by the reviewers follow on the next pages. We combined both replies into one document.

With best regards,

Dirk Görlich

Changes to the manuscript

- As requested, we added an eTOC summary and a running title.
- As requested by the Editor and the Reviewer 1, we have expanded the methods section and now included details of expression and purification of proteins as well as details of the assays demonstrated in Fig 3D and Fig 6AB.
- As requested by the Editor, conflict of interest and author contribution statements are included in to the Acknowledgements section and a summary paragraph for all supplemental material is added at the end of Methods section.
- As requested by Reviewer 1, we include a magnified image to Fig 4C, which demonstrates incompatibility of RanGTP-Ubc9 binding.
- As requested by the Reviewer 1, we have modified figure legends of Fig. 2E and 7C and included the surface potential values (+ and – 10 kcal/mol).
- As requested by the Reviewer 1, yellow letterings in Fig. S2 are replaced with darker shades.
- As requested by the Reviewer 1, the crystallographic table has been modified accordingly and missing information is included.
- We introduced a number of minor changes in the text to accommodate queries of the reviewers. These are detailed in the point-by-point reply below.

Reviewer #1

Aksu et al. describe the structural basis of nucleocytoplasmic transport mediated by the *S. cerevisiae* karyopherin Pdr6. In the accompanying manuscript, the authors identified the SUMO-conjugating enzyme UBC9 and the translation factor eIF5A as nuclear import and export cargo of Pdr6, respectively. To provide detailed insight into the mechanism of cargo binding and release, the authors determined the crystal structures of the Pdr6•RanGTP, Pdr6•UBC9, and the Pdr6•RanGTP•eIF5A complexes to a resolution of 2.9 Å, 4.53 Å, and 3.7 Å, respectively. Whereas the Pdr6•RanGTP structure closely resembles the binding mode of its mammalian homolog Imp13•RanGTP, the complexes of Pdr6•UBC9 and Pdr6•RanGTP•eIF5A show striking differences to their mammalian counterparts. Both UBC9 and eIF5A bind Pdr6 at entirely different interaction interfaces compared to the mammalian Imp13•UBC9 and Xpo4•RanGTP•eIF5A complexes. Interestingly, Aksu et al. demonstrate biochemically that *S. cerevisiae* and human eIF5A can form chimeric complexes with *S. cerevisiae* Pdr6 and its mammalian homolog Xpo4. In line with their structural studies, a mutational analysis shows that eIF5A indeed binds to different regions on Pdr6 and Xpo4. The authors then conclude that nuclear import of UBC9 and nuclear exclusion of eIF5A evolved more than once and independent from each other. Overall, the presented manuscript is a well conducted interdisciplinary study, with a very high technical quality as expected from the Görlich group. The determined crystal structures constitute an important advance in the field. However, to make the manuscript more accessible to the broad readership of JCB, the authors should address the following specific points:

Thank you!

1. In the abstract the authors state: "This suggests that non-canonical import of UBC9 and active nuclear exclusion [...]" What is the term "non-canonical" based on? To date, the only (published) available insight in the UBC9-karyopherin interaction is based on the UBC9•Imp13 structure. It would be beneficial to clarify the term given that, to the reviewer's knowledge, there is only one counter-example of UBC9 import in the literature.

Canonical import is defined on page 4: "Classic or canonical nuclear localization signals (NLSs) also function as linear motifs. They comprise either one or two short clusters of basic residues that dock into cognate binding-pockets of the nuclear import adapter importin α , which in turn uses importin β as the

actual transport receptor". To avoid confusion, we deleted the term from the abstract, because there, it is not yet defined.

2. The authors claim that "Pdr6 bound human Ran and *S. cerevisiae* Gsp1p equally well [...]." Do the authors show this anywhere? What are the dissociation constants? Either quantitative data should be included, or the text should be revised accordingly.

We did not determine dissociation constants, however, human Ran and yeast Gsp1 behave indistinguishably in binding assays with Pdr6 (the figure below is an example). To avoid the impression that we had measured affinities, we have deleted the quoted sentence in the revised manuscript.

3. "Pdr6 is an all alpha-helical protein built of 19 canonical HEAT repeats followed by three helices (termed HEAT 20) at its very C-terminus (Fig. 2A-D)." The C-terminus of a protein is the last amino acid, terminated by a carboxyl group. The authors should refer in their description maybe to "its very C-terminal region".

We changed the statement as suggested to "its very C-terminal region"

4. "Our Pdr6•RanGTP structure is indeed incompatible with Ubc9-binding (Fig.4)." Whereas the data are unambiguous, the visual representation in Figure 4C is not particularly meaningful. It would be better to highlight the clash with a magnified view to make the incompatibility of Ubc9 and RanGTP more accessible to the reader.

We included a magnified image to Fig. 4C to demonstrate the RanGTP-UBC9 clash.

5. "As discussed below, we assume that Ran puts the superhelical structure under tension, which could explain why the binary RanGTP Pdr6 interaction is only of moderate affinity (~230 nM [...]." It would be informative to compare this with the affinities of other RanGTP-NTR interactions.

We expanded this statement as follows: "... we assume that Ran has to put the super-helical structure under tension, which could explain why the binary RanGTP-Pdr6 interaction is of only moderate affinity (~230 nM; Hahn and Schlenstedt, 2011) even though a rather larger interface of 3876 Å² is buried. For comparison, importin β binds RanGTP with ~ 1 nM affinity (Bischoff and Görlich, 1997)."

6. "The N-terminal Ran-binding site is conserved in all so far analyzed importin beta family members. The C-terminal one, however, reflects an adaptation to specific binding modes of cargo binding and release." The authors should include a figure panel to make this conclusion more accessible to the reader.

Agreed. We included another figure to the supplements (now Fig. S3) It summarizes how many contacts Ran makes to individual HEAT repeats.

The figure shows that only the Ran-contacts of HEATs 1-3 are conserved. We also included sequence alignment of this region of NTRs. Although sequence conservation rather poor among NTRs, the positional conservation of RanGTP-interacting residues is high.

7. "The two recognize conserved features on this cargo - to the extent that Pdr6 binds yeast and human eIF5A equally well (Fig. 6A)." Figure 6A shows that Pdr6 binds human and yeast eIF5A, but does not allow for any quantitative statements. The current phrasing suggests that human and yeast eIF5A bind with similar affinities, which has not been tested. The authors should either revise their statement or provide quantitative data.

We rephrased this as follows: "Both, mammalian Xpo4 and yeast Pdr6, mediate nuclear export of eIF5A, both bind eIF5A from either yeast or human (Fig. 6A) and thus recognize conserved features on this cargo."

8. "First, Ran forms a common interaction interface with the cargo and the released binding energy promotes recruitment of the other NTR-ligand [...]." This sentence is confusing and requires clarification.

We have rephrased this. In context, it now reads:

Such switch can occur by two (not mutually exclusive) principles. First, Ran promotes cargo-binding to the exporter by contributing part of the cargo-binding interface. This applies to export of importin α , tRNA or pre-miRNAs. (Matsuura and Stewart, 2004; Cook et al., 2009; Okada et al., 2009). Second, Ran stabilizes the exportin or biportin in a high-energy (spring-loaded) 'nuclear conformation' with a then active cargo-binding site; This applies to CRM1 and Xpo4 (Monecke et al., 2009; Aksu et al., 2016).

9. Details for the experiments shown in Figure 6A and 6B are missing in the method section and should be added.

Thank you for pointing this out. We also missed the details of Figure 3D. We now included details for three experiments.

10. Figure 2E and 7C depict electrostatic surface potentials. The extreme points of the scale need to be indicated, or at least mentioned in the figure legend.

The extreme points of the scale are ± 10 kcal/mol. This information is now included in the figure legends of both figures.

11. Figure S1 visualizes the interaction of Pdr6 and Ubc9 with a magnified view in panel A that highlights salt bridges. In the absence of showing electron density in a separate figure panel, the accuracy of the depicted salt bridges is questionable at a resolution of 4.53 Å. At this somewhat low resolution I would not expect such a salt bridge to be well resolved.

We replaced the panel. It is the same view as before but without depicting the salt bridges.

12. The yellow residue labels in Figure S2 are basically unreadable and should be revised.

We used darker shades of yellow to make this figure more easily to read.

13. Crystallographic table, data and refinement statistics and structure quality. Unit cell dimensions and R factors should be rounded to one decimal and B factors should be rounded to full integers.

These numbers have been rounded as suggested.

Rmerge values and Rpim values (currently missing) should be included for all datasets and reported in percentages.

Implemented as suggested.

The high-resolution cutoffs should be determined with the I/σ in the highest resolution shell. While modern Pilatus detectors allow the usage of all data, an I/σ in the highest resolution shell of <1 is actually less data than background (1). Thus, the high-resolution cutoff should be adjusted to have an I/σ in the highest resolution shell of > 1 .

For the highest-resolution cutoffs, we chose shells, which still fulfill the $CC1/2 > 50\%$ and $I/\sigma > 1$ criteria.

$CC1/2$ values (currently missing) should be included for all datasets as well.

$CC1/2$ values have been included in the Table.

One of the datasets has an unusual multiplicity of ~ 400 . Is this really correct?

It is. We used a total of 20 454 618 reflections (60-3 Å) divided by 51 160 unique ones. The high multiplicity resulted from merging seven datasets (for SAD-phasing), from collecting data for two

complete rotations (720°) of each crystal, and from the high symmetric of this particular crystal form (#182; space group $P6_322$).

Ramachandran outliers should be 0%, especially for low resolution crystal structures as in this case.

With 4.5 Å resolution, it is clear that the structural model will not be perfect. Ramachandran outliers are one facet of that. We wish to point out, however, that our structure has actually less outliers (0.5%) than the average of structures of the same resolution.

Thus, published 4.5 Å crystal structures usually do have outliers. How many there are depends on the specific case and on how the constraints during the refinement have been weighted. Rectifying a persistent outlier would make it disappear from the statistics, however, it does not necessarily mean that its conformation is then modelled correctly. We discussed this issue with several colleagues, and there was actually agreement that overweighting the Ramachandran constraint would not be best practice, because it would invalidate Ramachandran-plot based validation.

Please note that our outliers were 'inherited' from the better resolved Pdr6·RanGTP structure that served as a molecular replacement model. They persisted during refinement of the Ubc9·Pdr6 structure.

Four of the six outliers are very close to the generously allowed regions (see below). Furthermore, most outliers reside in loops with conformational flexibility, where modelling runs into the problem that the best fitting *average structure* can represent a conformation that violates the Ramachandran rules. This problem can only be solved by modelling the structure as an ensemble.

None of the six outliers in our structure is close to any relevant interface, and none of them affects any conclusion that we draw.

MolProbity Ramachandran analysis

D_1200013365_model-annotate_P1.pdb, model 1

93.6% (1075/1148) of all residues were in favored (98%) regions.
99.5% (1142/1148) of all residues were in allowed (>99.8%) regions.

There were 6 outliers (phi, psi):
A 21 HIS (4.9, -78.2)
A 37 GLU (-38.9, -18.4)
A 414 PHE (54.5, 94.4)
A 419 VAL (23.0, -58.5)
A 693 PRO (-64.0, 34.3)
A 988 SER (-70.2, 30.7)

<http://kinemage.biochem.duke.edu>

Lovell, Davis, et al. Proteins 50:437 (2003)

14. English and grammar. The manuscript would benefit from substantial editing by the senior author.
The manuscript has been proof-read and corrected once again

Reviewer #2

The two manuscripts by Asku et al and Vera-Rodriguez et al define yeast Prd6 as a bidirectional NTR and provide the structural basis for its cargo recognition. Each of the two manuscripts is experimentally very extensive and taken together they define novel yeast export pathway that was previously unknown.

Major comments on Asku et al:

The authors have solved structures of key intermediates of the Prd6 transport cycle, specifically of ubc9-Prd6 import complex, RanGTP-Prd6 without cargo and RanGTP-Prd6-eIF5A export complex. The structures show some differences to their metazoan counterparts, that is the respective Xpo4 and Imp13 structures. These structures reveal the molecular basis for both Prd6-dependent pathways, as defined in the accompanying paper, in very high detail. The unconventional, bidirectional transport scheme of these pathways make this a very interesting paper, that does considerably go beyond our present structural knowledge of NTRs and their substrate binding mode. The fact that Prd6 also covers key interfaces of its substrates suggesting a chaperoning function is highly interesting! On a general note, solving as many cargo-NTR complexes as possible is important for this field. Since there is such a huge number of them, the existing structural repertoire is still quite limited.

I most enthusiastically recommend this paper for publication in JCB after minor revisions.

Thank you!

Minor comments on Asku et al:

- Page 6: Please make clear in the main text if the 'complex of full-length Pdr6 and the GTPase-deficient RanQ69L' contains yeast or human Ran. Does not matter much since its super-conserved, but should be transparent. Likewise, please clarify for Prd6-RanGTP complex.

We used human RanGTP throughout this paper, also for the Pdr6-RanGTP structures.

Page 7: Regarding the statement 'Furthermore, we found that Pdr6 imports UBC9, exports eIF5A and thus combines Imp13 and Xpo4 functions. This could suggest that Imp13 and Xpo4 evolved by gene duplication and diversification from an ancestral Pdr6 gene.' If you first say 'gene duplication and diversification', you make one assume that the ubc9 and eIF5A binding function existed before gene duplication, which would be inconsistent with that function having 'appeared twice in evolution'. What you mean to say is that most likely an ancient NTR gene was duplicated and the resulting genes adopted ubc9 and eIF5A binding functions independently, correct? Please consider making this clearer. On another note, the simplest imaginable evolutionary trajectory is not necessarily the most likely. In principle it could be both, convergent or divergent evolution, right?

The phrasing: "*This could suggest that Imp13 and Xpo4 evolved by gene duplication and diversification from an ancestral Pdr6 gene.*" was part of the introduction just before summarizing the results of this study. It was a rather plausible hypothesis, which got, however, contradicted by the new Pdr6 structures. To avoid confusion, we deleted the sentence from the introduction.

Comment on both papers:

Why would Phil1, Lsp1, eIF5A and eEF2 be exported by a different pathway than crm1-dependent export?! The authors suggest that an NES might interfere with the activity of a 'highly optimized translation factor', however, this argument does not apply to all Prd6 cargos. Could this have to do with the proposed nuclear chaperoning function of i.e. the hypusine-containing interface? But what is the evidence really that exposing this interface in the nucleus is harmful? Can one experimentally test this idea? Would exposing this interface in the nucleus by overexpressing eIF5A fused to a classical NLS (on top of the endogenous protein) cause a reduced fitness phenotype that depends on the expression level? Likewise, several other hypotheses spelled out in the discussion of mainly Asku et al, could well be tested. In more general terms, the strength of both papers is that they provide a very solid base to test the physiological relevance of the Prd6-dependent pathways. The two papers however do not yet capitalize on this capacity, which would be quite interesting because of the relevance for metazoan Xpo4 and Imp13. Both papers however already comprise an experimental tour de force. Extending the scope even further is certainly not required for publication.

These are all intriguing questions though they are not straightforward to answer. It is known that deleting Pdr6 impairs the fitness of the yeast (see e.g. *Qian et al., (2012). The genomic landscape and evolutionary resolution of antagonistic pleiotropy in yeast. Cell Rep. 2:1399-1410*). This loss of fitness will be the sum of several transport/ chaperone defects; we just do not know which individual defect has what impact on fitness. Ideally, one would construct Pdr6 mutant strains, where just one cargo is affected. However, without knowing the structures of all Pdr6·cargo complexes, this is impossible to do cleanly. Also, there might be overlaps of binding sites, and it might well be that import and export are coupled to each other – given that the transporter is a biportin.

The suggested eIF5A-NLS experiment is also complicated by a number of circumstances. First, the NLS contributes another basic patch, which is likely to alter its binding properties. It is not even clear if eIF5A can be fused to any peptide without losing activity (note that the GFP-fusion experiments in the accompanying paper were only possible because *S. cerevisiae* has two eIF5A copies). Second, eIF5A requires two enzymes for hypusination (deoxyhypusine synthase and deoxyhypusine hydroxylase) that are present in just limiting amounts. Overexpression will actually lead to under-hypusinated species (also of the remaining wild type proteins) and might cause a fitness defect. Sure, one could also over-express the modifying enzymes. However, our feeling is that all this might have a phenotype already.

Reviewer #3

This manuscript describes crystal structure analysis of yeast UBC9/Pdr6 import complex, RanGTP/Pdr6 heterodimer and RanGTP/Pdr6/eIF5A export complex. UBC9 and eIF5A are the new cargoes that was identified by the same authors' group in the accompanying paper. These yeast structures were compared with crystal structures of mammalian UBC9/Imp13 import complex and RanGTP/eIF5A/Exp4 export complex that was previously solved. Recognition of UBC9 by yeast Pdr6 differs from that of UBC9 by mammalian Imp13, and yeast eIF5 by Pdr6 from that of mammalian eIF5 by Exp4. The authors discuss evolutionary aspects on cargo recognition by nuclear transport receptors (NTRs).

Regarding cargo recognition, how the diverse cargos can be recognized by the same NTR is an intriguing problem. At the same time, evolutionary aspects of cargo recognition by NTR is also important untouched subject. The manuscript is well written and data are clear. I support aim of this study.

Thank you!

I have one question,

Fig6A, why yeast eIF5A do not bind to Pdr6 and Xpo4?

Unfortunately, yeast eIF5A and human Ran (5-180) run similarly in SDS-PAGE and hence bands partially overlap. In Fig. 6A, yeast eIF5A binds both Xpo4 and Pdr6. To clarify this point for the reader, we included the following statement to Fig 6 legend: “Please note that yeast eIF5A and Ran bands partially overlap in both panels.” Thanks for pointing this out.